# The Therapeutic Effects and Pathophysiology of Botulinum Toxin A on Voiding Dysfunction Due to Urethral Sphincter Dysfunction

**DOI:** 10.3390/toxins11120728

**Published:** 2019-12-13

**Authors:** Yao-Lin Kao, Kuan-Hsun Huang, Hann-Chorng Kuo, Yin-Chien Ou

**Affiliations:** 1Department of Urology, National Cheng Kung University Hospital, College of Medicine, National Cheng Kung University, Tainan 704, Taiwan; pleasewaitforg@hotmail.com; 2Department of Urology, Dalin Tzu Chi Hospital, Buddhist Tzu Chi Medical Foundation, Chiayi County 622, Taiwan; i5490108@hotmail.com; 3Department of Urology, Hualien Tzu Chi Hospital, Buddhist Tzu Chi Medical Foundation and Tzu Chi University, Hualien 970, Taiwan; hck@tzuchi.com.tw; 4College of Medicine, Institute of Clinical Medicine, National Cheng Kung University, Tainan 704, Taiwan

**Keywords:** botulinum toxin, urethral sphincter, urethral sphincter dysfunction, lower urinary tract symptoms, urodynamics

## Abstract

Neurogenic and non-neurogenic urethral sphincter dysfunction are common causes of voiding dysfunction. Injections of botulinum toxin A (BoNT-A) into the urethral sphincter have been used to treat urethral sphincter dysfunction (USD) refractory to conventional treatment. Since its first use for patients with detrusor sphincter dyssynergia in 1988, BoNT-A has been applied to various causes of USD, including dysfunctional voiding, Fowler’s syndrome, and poor relaxation of the external urethral sphincter. BoNT-A is believed to decrease urethral resistance via paralysis of the striated sphincter muscle through inhibition of acetylcholine release in the neuromuscular junction. Recovery of detrusor function in patients with detrusor underactivity combined with a hyperactive sphincter also suggested the potential neuromodulation effect of sphincteric BoNT-A injection. A large proportion of patients with different causes of USD report significant improvement in voiding after sphincteric BoNT-A injections. However, patient satisfaction might not increase with an improvement in the symptoms because of concomitant side effects including exacerbated incontinence, urinary urgency, and over-expectation. Nonetheless, in terms of efficacy and safety, BoNT-A is still a reasonable option for refractory voiding function. To date, studies focusing on urethral sphincter BoNT-A injections have been limited to the heterogeneous etiologies of USD. Further well-designed studies are thus needed.

## 1. Introduction

Urethral sphincter dysfunction (USD) is one of the functional causes of voiding dysfunction (VD), which leads to slow or incomplete micturition in both males and females [1]. The condition can be either neurogenic or non-neurogenic. While detrusor sphincter dyssynergia (DSD) commonly stands for the well-known neurogenic cause [2], dysfunctional voiding (DV), Fowler’s syndrome (FS), and poor relaxation of the external urethral sphincter (PRES) during voiding comprise the non-neurogenic causes [3,4]. Management of these functional disorders can be challenging if conventional treatment fails.

The introduction of a botulinum toxin A (BoNT-A) injection at the external urethral sphincter (EUS) was first performed in 1988 by Dykstra et al. [5]. Paralysis of the urethral sphincter and decreased urethral resistance were anticipated following the BoNT-A injection. The authors indeed found the signs of sphincter denervation and an improvement in voiding efficiency in patients with spinal cord injury (SCI) and DSD. Since then, extended use of BoNT-A in various urinary tract dysfunctions has been reported [6], however, the currently approved indications for BoNT-A for the lower urinary tract are neurogenic detrusor overactivity and overactive bladder [7]. To date, there have been only a handful of studies demonstrating the application of BoNT-A injections into the EUS, especially in the case of VDs other than DSD [8,9]. This review is an attempt to summarize the pathophysiology of BoNT-A and its therapeutic effects among different types of USD.

## 2. Biology and Mechanism of BoNT-A

Botulinum toxin is a neurotoxin produced by Clostridium botulinum [6]. It was first isolated as a crystalline product in 1946 [10] and was found to have a paralyzing effect on hyperactive muscles in the 1950s [11,12]. To date, eight immunologically antigenic distinct subtypes have been identified, i.e., subtypes A, B, C1, C2, D, E, F, and G [13]. Botulinum neurotoxin subtype A is potent, with the longest duration of action among these subtypes, and is commonly used in clinical practice [14]. Currently, several commercial forms of BoNT-A are available, where Botox^®^ (Allergan, Irvine, CA, USA) and Dysport^®^ (Ipsen, Slough, UK) are the two most widely used agents [15]. It should be noted that these products cannot be considered equivalent in terms of dose, efficacy, or safety owing to having different fragments of botulinum toxin and different manufacturing processes [6]. Although a few studies have suggested the efficacy of 1 unit of Botox to be similar to 3–5 units of Dysport [16,17], this simple linear exchange equation has been questioned in different BoNT-A applications [18].

The mechanism by which BoNT-A inhibits target muscle contractions is well-known as the blocking of acetylcholine release from presynaptic efferent nerves at the neuromuscular junctions via cleaving of synaptosome-associated protein 25 (SNAP-25) and preventing the docking of acetylcholine-containing vesicles to the neuronal cell membrane [19]. The toxin also blocks the release of other neurotransmitters, including adenosine triphosphate, substance P, calcitonin gene-related peptide, and downregulates sensory receptors such as vanilloid (TRPV1) and purinergic (P2 × 3) receptors [20,21] known as an afferent nerve desensitizer. The effect of BoNT-A on urethral striated muscle is thought to block the presynaptic release of acetylcholine in the neuromuscular junction and subsequently achieve a chemical sphincterotomy, which is believed to relieve the USD and improve VD [22,23]. An animal study has also shown a reduction in the release of norepinephrine in the urethra after BoNT-A injections at the EUS of rats, supporting its use in the treatment of EUS overactivity [24]. A decrease in maximum urethral pressure (MUP) of an average of 27 cm H_2_O was noted after BoNT-A injection in DSD patients [5].

Besides the direct effect on the EUS, BoNT-A injections at the EUS may lead to recovery of bladder detrusor contractility in VD patients with detrusor underactivity (DU) and hyperactive EUS or PRES [25]. It has been proposed that the suppression of EUS contraction will deactivate the afferent signals inhibiting the bladder reflex [26,27], however, this neuromodulation effect still awaits further confirmation.

## 3. Urethral Sphincter BoNT-A Injections in Patients with Detrusor Sphincter Dyssynergia

DSD is defined as involuntary contractions of the EUS during detrusor contractions [28]. DSD is mainly caused by damage to the upper motor neurons lying between the pontine micturition center and the sacral spinal cord, such as in the morbidities associated with spinal cord injury (SCI), multiple sclerosis (MS), and transverse myelitis (TM) [29,30,31,32,33]. DSD usually causes significant VD, leading to chronic urinary retention, recurrent urinary tract infection (UTI), high detrusor voiding pressure, autonomic dysreflexia, vesicoureteral reflux, and possible renal damage [34]. Alpha-blockers have been used to decrease outlet resistance, but the results of urodynamic studies have not been convincing [35]. Clean intermittent catheterization (CIC) is an effective alternative for patients to empty their bladder, but some patients are not able to tolerate it well due to upper limb impairment or psychological unwillingness [36]. Surgical sphincterotomy is another drastic option for those who fail in the treatments mentioned above, however, many of such patients experience a worsened quality of life due to persistent incontinence and a high long-term failure rate [37]. BoNT-A is thought to be an alternative, as it can block the release of acetylcholine from presynaptic vesicles at the neuromuscular junction, which causes temporary, reversible chemo-denervation of the external EUS [38].

Dykstra et al. [5] first described both transperineal and transurethral injections of BoNT-A to the EUS in 1988 as a new therapeutic approach for 11 male patients with SCI and DSD. In this preliminary study, weekly injections using different dosages of BoNT-A effectively decreased the MUP and post-void residual (PVR) urine volume. Electromyography (EMG) further confirmed sphincter denervation in all patients. The authors then conducted a small sample, randomized, double-blind placebo-control trial in five SCI male patients with DSD to compare low dose BoNT-A with normal saline by weekly injections for three weeks. They concluded that the therapeutic effects were found only in the BoNT-A injection group [39].

Instead of a once-weekly injection, a once-monthly dose has also been confirmed to effectively improve sphincter function as well as to decrease PVR [40]. In addition, a single transurethral dose of BoNT-A was further evaluated by Petit et al. [41] in 17 male patients with SCI and DSD. Their results showed similar clinical improvements in approximately 70% of the cases with an average therapeutic effect of two to six months. Phelan et al. [42] first confirmed the effectiveness of sphincteric BoNT-A injections in both male and female patients with various etiologies of DSD, including SCI, TM, and MS. De Sèze et al. [43] carried out a randomized, double-blind control trial to compare the efficacy of and tolerability to BoNT-A with a lidocaine single transperineal injection in patients with DSD. Higher patient satisfaction scores and significant decreases in PVR, MUP, and EMG activity were found only in the BoNT-A group. Most of the patients tolerated the treatment well without major complications.

Recently, more data describing the clinical experiences related to sphincteric BoNT-A injections in patients with DSD have been reported. Both transperineal injections [5,40,43,44,45,46,47] and transurethral injections using cystoscopy [5,40,41,42,48,49,50,51,52,53] showed promising outcomes in terms of reducing sphincteric activity. Although transurethral injections have been reported to be more effective in terms of decreasing the MUP than the use of a transperineal route [40], no further trials have ever directly compared these two injection methods. Most of the recent publications on this topic involve the use of a sphincteric injection of 100 units of Botox in DSD patients since a more prolonged therapeutic effect was found, as compared to 50 units, in a previous randomized control trial [54].

Urodynamic parameters were commonly used as measurements of objective outcomes after sphincteric BoNT-A injections in patients with DSD. As expected, a significant reduction in EMG activity [5,39,43] and a decrease in MUP [5,40,41,43,44] during the voiding phase have been found. Improvements in PVR [5,40,41,42,43,49], detrusor contraction pressure [5,44,48,49], and maximal flow rate (Qmax) [42,48] were also reported as a result of decreased sphincteric resistance. However, unlike SCI patients, a randomized, double-blind, placebo-control trial in 86 patients with MS and DSD showed that a single injection of BoNT-A did not decrease the PVR [44]. The authors posited that the unchanged PVR could be attributed to the lower baseline detrusor contraction pressure. In addition to the outcome measurement, urodynamic parameters have also been used to predict therapeutic outcomes. Several pre-treatment urodynamic parameters, including higher baseline detrusor contraction pressure [41,53], lower baseline sphincteric tone [48,50,55], and a synergic bladder neck [53,55] have been confirmed as predictors of a favorable outcome.

Although objective urodynamic results have been improved after EUS BoNT-A injection, Kuo et al. [52] reported inconsistencies between urodynamic outcomes and patient satisfaction. Even though the PVR and detrusor contraction pressure were improved, patients were not satisfied with the outcomes mainly because of the increased incontinence grade [51]. Notably, for patients with SCI and DSD, a detrusor BoNT-A injection provided a much better quality of life than the case with an EUS injection [52] This result emphasized the importance of continence in a patient’s quality of life. Other causes of patient dissatisfaction included persistent difficult urination [51], increases in urgency episodes [51], and the need for repeated injections [45]. For therapeutic outcomes, patient satisfaction was mainly due to improved voiding conditions and fewer autonomic dysreflexia (AD) episodes [52,53]. To summarize the subjective outcomes of sphincteric BoNT-A injections for patients with DSD, 61–88% experienced clinical improvement [5,40,41,42,48,51] for a therapeutic duration lasting two to six months [40,41,42,44], more than 80% could regain spontaneous voiding and successfully remove the indwelling catheter or stop CIC [42,48,49], and episodes of autonomic dysreflexia were reduced in more than 50% of the cases [5,43,50,51]. Table 1 summarizes the clinical studies on efficacy and adverse events related to sphincteric BoNT-A injections in the treatment of patients with DSD.

In spite of the benefits of sphincteric BoNT-A injections in patients with DSD based on the literature, the use of different injection protocols among various etiologies makes intergroup comparisons difficult. Further randomized control trials with large case numbers focusing on a single etiology of DSD are necessary to evaluate the therapeutic impacts on both objective and subjective parameters, quality of life, duration of effect, and long-term durability after repetitive injections.

## 4. Urethral Sphincter BoNT-A Injections in Children with Dysfunctional Voiding

DV is characterized by an intermittent or fluctuating flow rate, owing to intermittent contractions of the periurethral striated muscles or pelvic floor muscles during voiding in neurologically normal patients [56]. In 1973, Hinman and Baumann first described the symptom complex including enuresis, daytime wetting, UTI, and upper tract dilatation in 14 boys without neurologic defects and suggested that the condition is a functional discoordination between detrusor contraction and external sphincter relaxation [57]. This syndrome was then described by other authors as Hinman syndrome, occult neuropathic bladder, non-neurogenic neurogenic bladder, learned voiding dysfunction, and dysfunctional voiding [58,59,60]. In children, the typical symptoms of DV include urinary incontinence, recurrent UTI, voiding difficulty, urinary retention, and hydronephrosis [61]. To establish the diagnosis, uroflowmetry with an EMG is required to confirm that a sudden change in flow rate in the form of a staccato or intermittent pattern is related to sphincter contraction. Also, a “spinning-top” urethra can also be seen in a video-urodynamic study (vUDS) or voiding cystourethrography, indicating discoordination of the EUS and detrusor contraction during voiding [56].

The conventional treatments for children with DV include non-pharmacological urotherapy [62,63] and alpha-blockers [64]. Since DV and DSD share similar pathophysiology in terms of abnormal sphincteric activity during voiding, applying BoNT-A to the EUS seems to be a reasonable therapeutic option. A sphincteric BoNT-A injection was first introduced as a novel treatment for children with DV by Steinhardt et al. [65], who successfully improved incontinence and recurrent UTI in a 7-year-old girl, and also demonstrated a marked improvement in the degree of urethral dilatation.

Several case series with small samples also discussed the therapeutic outcome of BoNT-A in children with DV who failed traditional urotherapy and medical management [66,67,68,69,70]. According to these data, 80–85% of patients showed improvement in daytime incontinence or enuresis [68,69], total dryness was found in 45–80% of patients after sphincteric BoNT-A injections [67,68,69,70], and approximately 45–75% of patients were free from recurrent UTI even without prophylaxis antibiotics [68,69,70]. A small case series reported by Mokhless et al. [66] revealed that nine children who were catheterized preoperatively experienced recovery of spontaneous voiding after sphincteric BoNT-A injections. In the case of urodynamic parameters, PVR improvement was found in most of the studies, and a flow pattern changed to bell-shaped curve was also reported [69,70]. Unlike the usual dose of 50 to 100 units of Botox in pediatric sphincter injections, Franco et al. [67] used a higher dose ranging from 200 to 300 units in 16 children with DV. They reported long-lasting improvements in PVR at six months, and the majority of their patients did not require repeated injections. The authors hypothesized that BoNT-A could block sensory feedback of overactive guarding reflex, making it possible to retrain these children to void appropriately. No acute complications, including nausea, dysphagia, respiratory distress, or paralysis, were found in any of these studies. Clinical studies of sphincteric BoNT-A injections for children with DV are summarized in Table 2.

Although the effects and safety of BoNT-A use in children with DV seem to be convincing, we should remember that all these study designs were nonrandomized, without controlled variables, and comprised small samples. Further better-designed trials with longer follow up are necessary to arrive at an accurate conclusion.

## 5. Urethral Sphincter BoNT-A Injections in Adults with Dysfunctional Voiding

The precise prevalence of DV in the adult population is still unknown. In a urodynamic database review of 1015 adults who were evaluated for voiding symptoms, around 2% could be defined as having DV using strict vUDS criteria [60]. Adult DV may come from persistent disease since childhood or adult-onset symptoms due to non-neurological etiologies [72]. Although adults and children with DV share similar characteristics and are defined similarly [56,73], the clinical characteristics of these two groups are quite different. Unlike children, adult patients typically present with obstructive symptoms, followed by frequency, nocturia, and urgency. Recurrent UTI and urinary incontinence are less prominent in adults [60].

Data discussing the therapeutic effect of sphincteric BoNT-A injections in adults with DV are limited and are mostly provided by Kuo and his colleagues [48,55,71]. In a prospective nonrandomized study without controlled variables, the authors performed sphincteric injections using 50 to 100 units of Botox in 20 adults with DV and reported a subjectively excellent outcome in 30% of the patients, where the remaining 70% showed improvement [48]. Liao and Kuo also reported an overall success rate of 86.7% in adults with DV by sphincteric injections with 50 to 100 units of Botox in a five-year retrospective review. DU with low abdominal straining pressure, spastic EUS, and bladder neck obstruction were the most common causes of treatment failure [55].

A randomized, double-blind, placebo-controlled study was conducted in 31 adults with DV to compare the therapeutic effect of 100 units of Botox with normal saline [71]. Even though the detrusor voiding pressure and voided volume were significantly improved in the BoNT-A group, there were no significant between-group differences in the subjective success rate. The author hypothesized that the local injection itself might have some unknown therapeutic effects on the relaxation of the EUS [71]. This concept is similar to the dry needling effect on myofascial trigger point pain, which can relax the actin-myosin bonds and normalize muscle tone [74]. Additional well-designed studies to enroll more adult patients with DV are necessary to elucidate the therapeutic effect of sphincteric BoNT-A injections, normal saline injections, or even the dry needle effect. Clinical studies of sphincteric BoNT-A injections for adults with DV are summarized in Table 2.

## 6. Urethral Sphincter BoNT-A Injections in Patients with Fowler Syndrome

Fowler’s syndrome (FS), a specific cause of unexplained urinary retention in young women, was first described by Fowler in 1986 [75]. The condition is characterized by EUS relaxation failure with unique components of complex repetitive discharges and decelerating bursts presented in concentric needle EMG [3]. The typical feature of FS in vUDS include a large bladder capacity, reduced bladder sensation in the storage phase, decreased or no detrusor contraction with a patent bladder neck, and narrowing in the midurethra with or without ballooning of the proximal urethra [76]. The decrease in sensation and motor function in the bladder were thought to be a result of abnormally strong urethra afferent activity, which inhibits the bladder afferent signals to reach the brain as a spinal ‘pro-continence’ mechanism [77]. These findings are different from the pattern of the typical pattern of high-pressure low-flow in DV also caused by involuntary EUS or pelvic floor muscle contraction during voiding [78]. Whether FS is a subgroup of DV or a totally different entity remains currently unanswered.

Few studies have evaluated the effect of BoNT-A on the management of FS [55,79,80]. The first study was performed by Fowler and colleagues in 1992, where six women with FS were enrolled [79]. Two hundred units of BoNT-A (Division of Biologics, Porton Down, Salisbury, UK) were given to one side of the EUS under EMG guiding via a hollow cannula electrode. No improvements in voiding function were noted in any of the patients. One patient even developed transient stress urinary incontinence. In 2007, Liao and Kuo also reported no restoration of efficient voiding in two patients suspected to have FS with high MUP lacking a typical abnormal needle EMG pattern after injections with 100 units of Botox in four to eight EUS sites [55]. However, decreases in MUP and abdominal voiding pressure by 20 to 25 cm H_2_O after injections were noted by vUDS during follow up. In contrast to the poor outcome in the aforementioned studies, a 10-patient open-level pilot study in 2016 did find promising outcomes in the management of FS using BoNT-A [80]. The injections were done with 1 mL 2% lidocaine on either side of the external urethral meatus, followed by 100 units of Botox equally divided on either side of the EUS under EMG guidance. Four of five women with complete urine retention could void spontaneously four weeks after injections. Seven of the 10 women stopped CIC ten weeks after injections. Significant improvement in the Qmax, PVR, International Prostate Symptom Score (IPSS), and urethral pressure profile were also noted at ten weeks. No serious adverse effects were reported. Clinical studies on sphincteric BoNT-A efficacy and adverse FS events are summarized in Table 3.

Due to the rarity of the disease, the difficulties associated with arriving at a definitive diagnosis that needs special equipment, the techniques required for performance, and interpretation of concentric needle EMG, these studies were all limited to a small number of patients without adequate control groups. Further large cohort studies are needed to validate these outcomes. The contradictory findings might be the result of different BoNT-A injection techniques or the different etiology behind this disease. Compared to sacral neuromodulation, BoNT-A urethra injections might serve as a less invasive, low resource, safer alternative to other methods used to treat this disease.

## 7. Urethral Sphincter BoNT-A Injections in Patients with Poor Relaxation of The External Urethral Sphincter

PRES as a diagnosis was first described by Kuo in 2000 and was determined based on non-relaxed surface EMG activity combined with a narrow membranous urethra during the voiding phase in the vUDS [82]. It was believed to have a different pathophysiology beyond prostatic obstruction or bladder neck dysfunction in non-neurogenic male voiding dysfunction refractory to alpha-blocker or transurethral resection of prostate [83,84]. The concept was further applied to non-neurogenic females with the same EMG findings and narrowing of the distal urethra in vUDS [85]. The cause of PRES was posited to be multifactorial, including learned habituation, pelvic floor hypertonicity, increased bladder sensitivity, or occult neuropathy [86]. However, the exact etiology responsible for the poor relaxation of the EUS or pelvic floor remains to be elucidated [82].

The cardinal symptoms of PRES are hesitancy, small urine caliber, and terminal dribbling with an IPSS voiding-to-storage subscore ratio 1 [82,87]. PRES is characterized by relatively small but stable bladder [88] and low-pressure low-flow during voiding phase [89], which is different from the typical high-pressure low-flow presentation in DV or extremely large, compliant bladder in FS. The prevalence rates were 12–20% [87,89,90] and 17.6% [4] in male and female non-neurogenic VD refractory to medication patients, respectively. The incidence increased in young males [89] in patients with bladder pain syndrome [88] and in idiopathic DU patients [91]. Sphincteric BoNT-A injections might provide chemo-denervation of the EUS by inhibition of acetylcholine release in the neuromuscular junction to relieve the USD in PRES [23].

The improvement rate in clinical or urodynamic parameters after injection of 100 units of BoNT-A in patients with PRES was reported to be 79 to 96% [48,55]. However, with a stricter definition of excellent outcome, only 42% of such patients had restored spontaneous voiding or had a 25% improvement in urodynamic parameters [48]. Great patient satisfaction was also reported to be approximately 47–52% [81]. It was concluded that the major predictive factors for a successful outcome were opening of the bladder neck and a higher baseline Qmax, but not the type of USD [81]. An increased recovery rate of detrusor contractility was further reported in idiopathic DU combined with PRES [25]. This result supported the hypothesis suggesting that the low-pressure low-flow dysfunction presented in PRES might be the result of detrusor suppression induced by non-relaxed EUS activity. With the aid of EUS relaxation after BoNT-A injections, the suppressed detrusor function was resumed. Clinical studies on sphincteric BoNT-A efficacy and adverse events of PRES are summarized in Table 3. Notably, most of the therapeutic effects of EUS BoNT-A injections in PRES came from studies conducted by Kuo’s research group. Further work from other clinical facilities and laboratories might lead to more prudent inferences.

## 8. Conclusions

In recent decades, BoNT-A has been used in the treatment of VD caused by various types of USD. It has been reported to be effective in the management of DSD, DV, PRES, and has shown promise in treating FS. However, patient satisfaction might not correlate well with objective improvement. BoNT-A injections may serve as a less invasive and safer option in treatment of USD refractory to conventional medications. The mechanism by which BoNT-A improves USD is thought to be a result of a decrease in urethral resistance via inhibition of acetylcholine released in the presynaptic neuron of the EUS, and through the recovery of detrusor muscle contractility via neuromodulation. Studies focused on BoNT-A injections at the EUS have often been limited to distinct etiologies of USD. Further well-designed clinical and basic studies are needed to confirm its effect.

## Figures and Tables

**Table 1 toxins-11-00728-t001:** Summary of clinical studies using sphincteric botulinum toxin A (BoNT-A) injections for patients with detrusor sphincter dyssynergia (DSD).

Author (Year)	Sex (No.)	Cause of DSD (No.)	Injection Method and Dose	UDS Improvements	Clinical Improvements (Events/Total Cases)	Adverse Events (Events/Total Cases)	Effective Duration
**Randomized control trials**							
Dykstra and Sidi (1990) [39]	M (5)	SCI (5)	Transurethral low dose BoNT-A, weekly Transurethral N/S, weekly	PVR, MUP, EMG activity ^a^	NA	Nil	NA
de Sèze et al. (2002) [43]	M (12) F (1)	SCI (9), MS (3), Congenital (1)	Transperineal 100U Botox Transperineal Lidocaine	PVR, MUP, EMG activity ^a^	Higher satisfaction score in the Botox group Voiding function improved in the Botox group Less AD (3/4) in the Botox group	Nil	3 months: 31% =3 months: 46% 3 months: 23%
Gallien et al. (2005) [44]	M (28) F (58)	MS (86)	Transperineal 100U Botox Transperineal N/S	MUP, Pdet, VV ^a^	No between-group differences No improvement in IPSS and VAS	UTI (16/45) Incontinence (2/45) Fecal incontinence (1/45)	2 months
Kuo (2007) [54]	M or F (66)	DSD (6), Non-DSD (60)	Transurethral (M) or periurethral (F) 50U Botox Transurethral (M) or periurethral (F) 100U Botox	PVR, MUP, Pdet, QmaX	Excellent outcome (5/6) for DSD patients No differences between 50U and 100U	Nil	50U: 6.4 months 100U: 8.4 months
**Nonrandomized control trial**							
Kuo (2013) [52]	M or F (55)	SCI (47), MS (6), TM (2)	Transurethral (M) or periurethral (F) 100U Botox Intradetrusor 200U Botox	PVR, Pdet, Qmax ^a^	Greater QoL improvement with detrusor injection than with sphincter injection	Incontinence is the major cause of dissatisfaction for sphincter injection	NA
**Non-control open label trials**							
Dykstra et al. (1988) [5]	M (11)	SCI (11)	Transperineal 20-80U BoNT-A, weekly Transurethral 80-240U BoNT-A, weekly	PVR, MUP, EMG activity	Less AD (5/7)	Nil	50 days
Schurch et al. (1996) [40]	M (24)	SCI (24)	Transperineal 250U Dysport, monthly Transurethral 100U Botox, monthly	PVR, MUP	Sphincter function improved (21/24)	Nil	3–9 months
Petit et al. (1998) [41]	M (17)	SCI (17)	Transurethral 150U Dysport	PVR, MUP, Pdet	Modality of voiding improved (10/17)	Urethral bleeding (1) Incontinence (5)	2–6 months
Phelan et al. (2001) [42]	M (8) F (13)	SCI (1), MS (9), TM (2), Non-DSD (9)	Transurethral 80-100U Botox	PVR, Qmax	Voiding pattern improved (14/21) Regain of spontaneous voiding (19/21)	Nil	3 months
Kuo (2003) [48]	M (48) F (55)	DSD (29), Non-DSD (74)	Transurethral (M) or periurethral (F) 100U Botox	Pdet, Qmax	Excellent outcome (8/29) Improved outcome (15/29)	NA ^b^	2–6 months
Smith et al. (2005) [49]	M or F (68)	SCI (9), MS (32), Non-DSD (27)	Transurethral 80-200U Botox	PVR, Pdet, Capacity	Regain of spontaneous voiding (34/41)	Incontinence (3/68)	6 months
Liao and Kuo (2007) [55]	M (112) F (88)	DSD (48), Non-DSD (152)	Transurethral (M) or periurethral (F) 50-100U Botox	NA	Excellent outcome (19/48) Improved outcome (26/48)	Nil	NA
Kuo (2008) [51]	M (22) F (11)	SCI (26), MS (5), TM (2)	Transurethral (M) or periurethral (F) 100U Botox	PVR, Pdet, Qmax	Improved IIQ-7 and UDI-6 Voiding function improved (26/33) Less AD (3/6)	Incontinence (16/33) Increase urgency (5/33) De novo frequency (3/33)	Patients received repeat injection at 4–9 months
Chen et al. (2008) [50]	M (17) F (3)	SCI (20)	Transurethral 100U Botox	MUP, EMG activity	Vesico-ureteral reflux resolved (1/1) Less AD (4/4)	Mild hematuria (2/20)	NA
Tsai et al. (2009) [46]	M (18)	SCI (18)	Transperineal 100U Botox	PVR, MUP, Pdet	Less symptomatic UTI (11/13) Modality of voiding improved (17/18) Hydronephrosis resolved (7/9) Vesico-ureteral reflux resolved (1/1) Less AD (6/7)	Nil	3 months
Chen et al. (2010) [47]	M (18)	SCI (18)	Transperineal 100U Botox	PVR, MUP, EMG activity	Less AD (5/5)	Mild hematuria (1/20)	2–6 months
Huang et al. (2016) [53]	65	SCI (65)	Intradetrusor 200U and transurethral 100U Botox	MUP, Pdet, VV	Urgency incontinence improved (59/59) Incontinence resolved (25/59) Less symptomatic UTI (6/14) Less AD (11/18)	Nil	NA
Soler et al. (2016) [45]	M (72) F (27)	SCI (99)	Transperineal 100U Botox	PVR	Excellent outcome (48/99) Modality of voiding improved (25/99) Vesico-ureteral reflux resolved (6/11) Less AD (69/82)	Nil	6.5 months

AD = Autonomic dysreflexia; BoNT-A = Botulinum toxin A; DSD = Detrusor sphincter dyssynergia; EMG = Electromyogram; F = Female; IIQ-7 = Incontinence Impact Questionnaire–Short Form; IPSS = International prostate symptom score; M = Male; MS = Multiple sclerosis; MUP = Maximal urethral pressure; NA = data not accessible from the study; Nil = none; No. = Number; N/S = Normal saline; Pdet = Detrusor contraction pressure; PVR = Post-void residual urine volume; Qmax = Maximal flow rate; QoL = Quality of life index; SCI = Spinal cord injury; TM = Transverse myelitis; UDI-6 = Urogenital Distress Inventory–Short Form; UDS = Urodynamic study; UTI = Urinary tract infection; VAS = Visual analog scale; VV = Voided volume. Sphincteric injections used with preparation other than the typical BoNT-A commercial form, including Botox or Disport, were denoted as “BoNT-A”. ^a^ UDS improvements were found in the urethral Botox group. ^b^ Individual results in specific disease groups were not available.

**Table 2 toxins-11-00728-t002:** Summary of clinical studies using sphincteric BoNT-A injections for children and adults with dysfunctional voiding (DV).

Author (Year)	Sex (No.)	Disease (No.)	Injection Method and Dose	UDS Improvements	Clinical Improvements (Events/Total Cases)	Adverse Events (Events/Total Cases)	Effective Duration
**Studies regarding BoNT-A injection in children DV**							
Mokhless et al. (2006) [66]	M (6) F (4)	DV (10)	Transurethral 50-100U Botox	PVR, Qmax, EMG activity	Regain of spontaneous voiding (9/9) Hydronephrosis resolved (2/4) Hydronephrosis downgraded (2/4) Vesico-ureteral reflux resolved (1/1)	Nil	6 months
Petronijevic et al. (2007) [70]	F (9)	DV (9)	Transperineal 500U Dysport	PVR, VV, voiding pattern	Improved voiding function (7/9) Incontinence resolved (4/5) Recurrent UTI resolved (6/8)	Nil	6 months
Franco et al. (2007) [67]	M or F (16)	DV (16)	Transurethral 200-300U Botox	PVR	Incontinence resolved (13/16) Recurrent epididymo-orchitis resolved (3/3)	Nil	6 months
Vricella et al. (2014) [68]	M (8) F (4)	DV (12)	Transurethral (M) or periurethral (F) 100U Botox	PVR, Qmax	Voiding condition improved (8/12) Incontinence resolved (4/7), improved (2/7) Hydronephrosis resolved (1/2) Vesico-ureteral reflux resolved (1/3) Recurrent UTI resolved (4/7) Discontinued anticholinergics (6/6)	Nil	Repeat injection at 6–21 months
’t Hoen et al. (2015) [69]	M (4) F (16)	DV (20)	Transurethral (M) or periurethral (F) 100U Botox	PVR, voiding pattern	Incontinence resolved (9/20), improved (7/20) Recurrent UTI resolved (5/11), improved (6/11)	Sudden increase of incontinence (9/20) Gluteus maximus muscle numbness (1/20)	Repeat injection after 13 months in average
**Studies regarding BoNT-A injection in adult DV**							
Kuo (2003) [48]	M (48) F (55)	DV (20) Non-DV (83)	Transurethral (M) or periurethral (F) 50-100U Botox	Pdet, Qmax	Excellent outcome (6/20) Improved outcome (14/20)	NA ^a^	2–6 months
Liao and Kuo (2007) [55]	M (112) F (88)	DV (60) Non-DV (140)	Transurethral (M) or periurethral (F) 50-100U Botox	NA	Excellent outcome (37/60) Improved outcome (15/60)	Nil	NA
Kuo (2007) ^b^ [54]	M or F (66)	DV (21) Non-DV (45)	Transurethral (M) or periurethral (F) 50U Botox Transurethral (M) or periurethral (F) 100U Botox	NA	Excellent outcome (13/21) for DV patients Improved outcome (6/21) for DV patients No difference between 50U and 100U	Nil	50U: 6.4 months 100U: 8.4 months
Jiang et al. (2016) ^b^ [71]	M or F (62)	DV (38) Non-DV (24)	Transurethral (M) or periurethral (F) 100U Botox Transurethral (M) or periurethral (F) N/S	Pdet, Qmax, VV ^c^	IPSS, QoL, and PPBC improved in both groups Success outcome (7/16) for Botox	De novo UUI (3/62) UTI (3/62) Micturition pain (2/62) Hematuria (2/62)	NA

BoNT-A = Botulinum toxin A; DV: dysfunctional voiding; EMG = Electromyogram; F = Female; IPSS = international prostate symptom score; M = Male; NA: data not accessible from the study; Nil = none; No. = Number; N/S = normal saline; Pdet = Detrusor contraction pressure; PPBC = Patient perception of bladder condition; PVR = Post-void residual urine volume; Qmax = Maximal flow rate; QoL = Quality of life index; UDS = Urodynamic study; UTI = Urinary tract infection; UUI = Urgency urinary incontinence; VV = Voided volume. ^a^ Individual results toward specific disease group were not assessable. ^b^ Both studies were designed as randomized control trials. ^c^ UDS improvements were found in the urethral Botox injection group for patients with DV.

**Table 3 toxins-11-00728-t003:** Summary of clinical studies using sphincteric BoNT-A injections for patients with Fowler’s syndrome (FS) and poor relaxation of the external urethral sphincter (PRES).

Author (Year)	Sex (No.)	Disease (No.)	Injection Method and Dose	UDS Improvements	Clinical Improvements (Events/Total Cases)	Adverse Events (Events/Total Cases)	Effective Duration
**Studies regarding BoNT-A injection in FS**							
Fowler et al. (1992) [79]	F (6)	FS (6)	Transperineal 200U BoNT-A	NA	No women restored normal micturition reflex	SUI (1/6)	NA
Liao and Kuo (2007) [55]	M (112) F (88)	FS (2) ^a^ Non-FS (198)	Transperineal 100U Botox	MUP	No improvement in voiding efficiency	Nil	NA
Panicker et al. (2016) [80]	F (10)	FS (10)	Transperineal 1 mL 2% lidocaine followed by 100U Botox	PVR, Qmax, MUP	IPSS improvement (8/10) Stopped CIC (7/10)	Nil	12–14 weeks
**Studies regarding BoNT-A injection in PRES**							
Kuo (2003) [48]	M (48) F (55)	PRES (19) Non-PRES (84)	Transurethral (M) or periurethral (F) 100U Botox	PVR	Excellent outcome (8/19) Improved outcome (7/19)	NA ^b^	2–6 months
Liao and Kuo (2007) [55]	M (112) F (88)	PRES (23) Non-PRES (177)	Transurethral (M) or periurethral (F) 100U Botox	NA	Excellent outcome (12/23) Improved outcome (10/23)	Nil	NA
Kuo (2007) [25]	M (22) F (5)	PRES (5) Non-PRES (22)	Transurethral (M) or periurethral (F) 50-100U Botox	PVR, Pdet, Qmax	Significant voiding and QoL improvement ^b^	Nil	NA ^b^
Lee et al. (2019) [81]	M or F (155)	PRES (17) Non-PRES (138)	Transurethral (M) or periurethral (F) 100U Botox	Voiding efficiency	Improved voiding efficiency and global response assessment (8/17)	NA ^b^	NA

BoNT-A = Botulinum toxin A; CIC = Clean intermittent catheterization; F = Female; FS = Fowler’s syndrome; IPSS = international prostate symptom score; M = Male; MUP = Maximal urethral pressure; NA = data not accessible from the study; Nil = none; No. = number; Pdet = Detrusor contraction pressure; PRES = Poor relaxation of the external urethral sphincter; PVR = Post-void residual urine volume; Qmax = Maximal flow rate; QoL = Quality of life index; SUI = Stress urinary incontinence; UDS = Urodynamic study. None of these studies were randomized or controlled. Sphincteric injections were given with preparation other than typical BoNT-A commercial form including Botox or Disport denoted as “BoNT-A”. ^a^ The subjects enrolled were not typical FS patients. They had a very high baseline MUP but did not have typical patterns of FS presented in a concentric needle electromyographic study. ^b^ Data were analyzed using combined groups. Individual results for specific disease groups were not available.

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
