# Peer review of "The Therapeutic Effects and Pathophysiology of Botulinum Toxin A on Voiding Dysfunction Due to Urethral Sphincter Dysfunction"

_toxins, 2019, doi:10.3390/toxins11120728_

Round 1
Reviewer 1 Report
This review articles mainly contains the effects of Botulinum toxin type A on various voiding dysfunction due to USD including cons and pros.
Still, this procedure has not generally approved, even though injection to the neurogenic detrusor overactivity has been approved. This article represents well current status of the clinical trials of Botulinum toxin type A on various voiding dysfunction due to USD.
Reviewer 2 Report
The authors reviewed the literature about the therapeutic effects and pathophysiology of BoNT-A on voiding dysfunction due to urethral sphincter dysfunction.
This manuscript is interesting and new, nevertheless, this paper need some necessary corrections before publishing.
Keywords
Please include also “urethral sphincter dysfunction.
Introduction
Line 39: please add references at the end of the sentence: ..”has been reported; ”
Line 41: please add references at the end of the sentence.
Line 45: please add references at the end of the sentence.
Line 51: please add references at the end of the sentence: “.. two most widely used agents”
Line 77: please add references at the end of the sentence ..(TM).
Table 1.
Please add to abbreviations: Nil; IIQ-7 and UDI-6.
Line 159: please add the year of the publication: Hinmann and Baumann (…?).
Please say in manuscript: Table 1; Table 2; Table 3.
Table 2.
Please add to abbreviations: Nil; Pdet.
Line 236: please add references at the end of the sentence: “.. needle EMG”..
Line 245: please add references at the end of the sentence.
Line 249: please add the year of the publication: Liao and Kuo (…?).
Line 251: please add references at the end of the sentence.
Table .
Please add to abbreviations: Nil.
Line 295: please add references at the end of the sentence.
Round 2
Reviewer 2 Report
This manuscript was suffizient corrected and improved.